# On the Reproducibility of: Improvement-Focused Causal Recourse

## Abstract

This work aims to reproduce the main findings of "Improvement-Focused Causal Recourse (ICR)"(König et al., 2023) within the field of algorithmic recourse recommendations. The authors demonstrate that acceptance-focused recourse recommendation methods, like counterfactual explanations (CE), may suggest actions that revert the model's verdict by gaming the predictor whenever lucrative. To tackle this, the authors introduce ICR, which focuses on improvement by optimizing for a new target variable in their causal model. It is also demonstrated that improvement guarantees consequently translate into acceptance guarantees. We can confirm the findings of the original paper. The contribution of the current study is a more extensive assessment of the robustness and generalizability of ICR. Various techniques were employed to test the algorithm's performance under different architectural choices, such as different classifiers or optimization methods, data and model shifts, and a new dataset. Our findings suggest that ICR is more robust than CE and causal recourse (CR).

## 1 Introduction

As the deployment of predictive systems becomes more and more prevalent in critical areas of decision-making such as employee hiring (Raghavan et al., 2020), organ transplant priority determination (Obermeyer & Mullainathan, 2019), or judiciary decisions (Zeng et al., 2017), more emphasis should be placed in algorithmic explainability methods that offer the explainee an intuitive understanding of the system, and the possibility to apply recourse, i.e. actions that revert an unfavorable decision. In the domain of algorithmic decision-making, recourse methods play a crucial role in informing stakeholders on actions to reverse unfavorable model predictions. Counterfactual explanations (CE) are concerned with changing the inputs to the model such that the model prediction changes in the desired way (Wachter et al., 2017). Karimi et al. (2021; 2022) recognize recourse to be a causal problem instead of a counterfactual one and proposes recourse through minimal interventions, emphasizing the actions that need to be taken to achieve a favorable decision. It utilizes causal knowledge to translate into recommendable recourse actions.

Traditional approaches, such as Causal Recourse (CR), primarily focus on achieving acceptance (reverting the model's decision) without necessarily ensuring improvement in the underlying real-world state. This emphasis on acceptance can lead to actions that may deceive the predictor without effectively addressing the actual improvement needed. To address this limitation, König et al. (2023) introduce Improvement-Focused Causal Recourse (ICR). In ICR, recommendations are explicitly geared towards achieving improvement and are not tailored for acceptance by a specific predictor. Causal knowledge is leveraged through structural causal models (SCMs) (Pearl et al., 2000) or causal graphs to make improvement-focused recommendations, and in the case where the SCM is known it can be utilised to design decision systems that accurately predict both pre- and post-recourse. In this work, we aim to reproduce the authors' findings, verify their claims, and perform additional experiments to assess the robustness and generalizability of the proposed method, providing further evidence to strengthen their claims. The code of our project is available here.

## 2 Scope of reproducibility

Improvement-Focused Causal Recourse belongs in the family of local post-hoc explainability methods. They are one of the more "human-friendly" approaches (Molnar, 2019) towards the goal of algorithmic transparency since they are contrastive to the current instance of a specific individual and selective since they focus on a small number of feature changes. ICR improves upon this by considering the causal dependencies with the real world state. The innovation of ICR is that, while CE and CR aim to revert the prediction, ICR seeks to revert the target, i.e., the underlying ground truth. The latter makes it a more holistic method for understanding the dynamics between input features and the outcomes in the context of algorithmic decision-making.

In the current reproducibility study, our main goal is to verify the following claims of the original paper:

- **Claim 1 - Attaining Improvement:** ICR reliably guides individuals towards actions that lead to improvement in scenarios where gaming is possible and lucrative.

- **Claim 2 - Attaining Acceptance:** CE, CR, and ICR all lead to acceptance, but CE and CR show higher observed acceptance rates than ICR.

- **Claim 3 - Attaining Acceptance Robustly:** ICR actions are more likely to be accepted by other model fits with similar performance on the same data.

- **Claim 4 - Recommendation Cost:** ICR actions are more costly than CR but lead to improvement, acceptance, and greater robustness to model refits.

In addition to reproducing the results presented in the paper, we perform additional experiments that test the robustness and, to some extent, the generalizability of the approach.

## 3 Background

### 3.1 Structural Causal Model

We define the SCM(Structural Causal Model) $\mathcal{M} \in \Pi$(Pearl, 2009) with $\mathcal{M} = (X, U, \mathbb{F})$. The SCM consists of endogenous (observed) variables $X \in \mathcal{X}$, exogenous variables $U \in \mathcal{U}$, and structural equations $\mathbb{F}$. The structural equations $\mathbb{F} : \mathcal{U} \to \mathcal{X}$ define how to obtain the endogenous variables given the exogenous variables. $\mathcal{M}$ is illustrated in a directed graphical model $\mathcal{G}$ (see, e.g., Appendix Fig. 3, where the exogenous variables are for simplicity not shown).
Pearl's ladder of causation (Pearl, 2009) further divides SCMs into three parts: rung 1 (observation): SCMs can describe (conditional) distributions; rung 2 (interventions): SCMs can predict the effect of actions do(x); rung 3 (counterfactuals): SCMs can imagine different outcomes if a different action would have been done. Actions are modeled via structural interventions $a : \Pi \to \Pi$, which are transformations on the SCM. The interventions are defined as $a = do(\{X_i := a_i\}_{i \in I})$. $I$ contains the indices of the intervened-upon variables. The do-operator substitutes the equation for $X_i$ in $\mathbb{F}$ as follows: $X_i := a_i$. Furthermore, all edges in the graph $\mathcal{G}$ leading to $X_i$ are removed. This creates the intervened upon structural model $\mathcal{M}_a$ with the equations $\mathbb{F}_a = \{F_i\}_{i \notin I} \cup \{X_i := a_i\}_{i \in I}$.
To calculate counterfactuals, the following 3-step procedure (Pearl, 2009) can be used, under the assumption that $\mathcal{M}$ is an additive noise model: Abduction: The distribution of exogenous variables $U$ is inferred from the observations $x^{pre}$; Action: the do(a) interventions are performed as described above; Prediction: We can sample from the counterfactual distribution using the intervened upon equations $\mathbb{F}_a$ and the inferred noise.

### 3.2 Counterfactual Explanations

To calculate counterfactual explanations (CE), the following formulas(Ustun et al., 2019) are used:

$$\delta^* \in \operatorname{argmin}_\delta \quad \operatorname{cost}\left(\delta, x^{\mathrm{pre}}\right) \quad \text{s.t.} \quad h(x^{\mathrm{CFE}}) \neq h(x^{\mathrm{pre}}) \tag{1}$$
$$x^{\mathrm{CFE}} = x^{\mathrm{pre}} + \delta$$
$$x^{\mathrm{CFE}} \in \mathcal{P}, \delta \in \mathcal{F}$$

Where $x^{\mathrm{pre}}$ are the factual values, cost is a user-defined function to calculate the cost of changing $x^{\mathrm{pre}}$ by $\delta$, $\mathcal{P}$ is a set of plausibility constraints, $\mathcal{F}$ is a set of feasibility constraints as defined by Karimi et al. (2021). This builds upon the assumption that $\delta^*$ contains the minimal actions needed to reverse the decision of an algorithm. This assumption does not always hold since changing some variables might have downstream effects, and thus, it might still reverse the decision, but it is not the minimal set of actions.

### 3.3 Causal Recourse

To tackle this issue, causal recourse (CR)(Karimi et al., 2021) was developed. It is assumed that an SCM is given to CR. This SCM might come from a field expert; however, in many cases, an SCMan SCM might not exist, and it is impossible to do so. To calculate CR, the following formulas are used:

$$a^* \in \operatorname{argmin}_a \quad \operatorname{cost}\left(a, x^{\mathrm{pre}}\right) \quad \text{s.t.} \quad h(x^{\mathrm{SCF}}) \neq h(x^{\mathrm{pre}}) \tag{2}$$
$$x^{\mathrm{SCF}} = \mathbb{F}_a(\mathbb{F}^{-1}(x^{\mathrm{pre}}))$$
$$x^{\mathrm{SCF}} \in \mathcal{P}, a \in \mathcal{F}$$

where $x^{\mathrm{SCF}}$ is the resulting structural counterfactual. This formula assumes there are no hidden confounding variables and a fully specified invertible for $\mathbb{F}$ exists such that $\mathbb{F}(\mathbb{F}^{-1}(x)) = x$. CR differs from CE because it considers the downstream effect that might exist by intervening upon a certain variable.

## 4 Improvement-Focused Causal Recourse

The ICR mechanism proposed by König et al. (2023) is one of the first recourse methods proposed that ensures reversion of the underlying real-world state (improvement) while also leading to acceptance (reverting an unfavorable decision). ICR utilizes the causal knowledge of an SCM, inspired by Karimi et al. (2021) to steer individuals who need recourse towards improvement. In many applications, an SCM might not exist due to the difficulty of inferring the relationships between different variables. However, the knowledge about a causal graph might exist, and ICR uses, in those cases, a subpopulation approach, similar to how Karimi et al. (2020) solved the issue of not having an SCM at hand.

### 4.1 Individualized Improvement Confidence

The authors define $\gamma^{\mathrm{ind}}$, which is the individualized improvement confidence as follows:

$$\gamma^{\mathrm{ind}}(a) = \gamma(a, x^{\mathrm{pre}}) := P(Y^{\mathrm{post}} = 1 | do(a), x^{\mathrm{pre}}) \tag{3}$$

for an action a and the datapoint $x^{\mathrm{pre}}$. Y is an introduced target variable that captures improvement. This target variable is part of the SCM. $Y^{\mathrm{post}}$ is the post recourse target.

### 4.2 Subpopulation Improvement Confidence

In the case where the SCM is not known, we assume no observed variables influence both the dependent and the independent variables. We have to fall back to the effect of interventions (rung 2 in Pearl's ladder of causation (Pearl, 2009)). Since the interventional distribution captures broader characteristics of the whole population, it cannot accurately capture the action's effects on specific individuals. In this scenario, a subpopulation-based improvement confidence expresses the probability of improvement $Y$ being a desired outcome in a subgroup of individuals with similar characteristics. This subgroup is created by choosing

individuals whose values in the variables unaffected by the action a ($G_a$) are similar. The author's definition of subpopulation improvement confidence is as follows:

$$\gamma^{\text{sub}}(a) = \gamma(a, x_{G_a}^{\text{pre}}) := P(Y^{\text{post}} = 1 | do(a), x_{G_a}^{\text{pre}}) \tag{4}$$

### 4.3 Optimization Problem

To generate ICR actions, the authors define Equation 5, which optimizes the cost for the actions. The objective is to discover actions that inflict a minimal cost while constrained by a user-specified improvement target confidence $\bar{\gamma}$. This confidence can be intuitively interpreted as the probability of improvement, given that the individual follows the recommended recourse actions. If we choose a higher confidence, the probability of recourse is higher. However, this might come with a higher cost associated with the user. At the same time, if we set a low target confidence (below 0.75), it is not worthwhile for the user to act on the recourse recommendation since the probability of recourse is barely higher than that of change. The cost function cost $(a, x^{\text{pre}})$ reflects the effort needed by an individual to complete an action a. The optimization objective for ICR can be interpreted as two smaller intervention objectives (Karimi et al., 2020). First, optimization is applied to the intervention targets $I_a$, followed by optimizing intervention values $\theta_a$. Considering our objective is to achieve improvement, we limit $I_a$ to all parents of $Y$. The authors motivate their decision to use the genetic algorithm NSGA-II(Deb et al., 2002) for optimizing the constrained objective below, following previous work (Dandl et al., 2020).

$$\text{argmin}_{a=do(X_I=\theta)} \quad \text{cost}(a, x^{\text{pre}}) \quad \text{s.t.} \quad \gamma(a) \geq \bar{\gamma} \tag{5}$$

### 4.4 Improvement leads to acceptance

Equation 5 optimizes actions for improvement, but this does not automatically lead to acceptance. Suppose an individualized pre-recourse predictor is used for post-recourse prediction. In that case, there is an imbalance in predictive power since ICR uses $x^{\text{pre}}$ and the SCM and the post-recourse prediction only uses the knowledge of $x^{\text{post}}$. The authors solve this by also utilizing the SCM for the post-recourse prediction by defining the following individualized post-recourse predictor:

$$h^{*,\text{ind}}(x^{\text{post}}) = P(Y^{\text{post}} = 1 | x^{\text{post}}, x^{\text{pre}}, do(a)) \tag{6}$$

For the subpopulation approach, the pre-recourse predictor remains accurate and does not have an imbalance in predictive power. This leads the authors to the interesting conclusion that CR can lead to improvement if it only acts upon causes of the underlying world-state Y.
The defined post-recourse predictors show that acting upon improvement also leads to acceptance. Therefore, by making the recourse recommendation, following the true underlying world state, the predictor will also accept this recourse with a certain confidence.

## 5 Methodology

This section breaks down our approach to confirming the original study's findings and further investigates ICR capabilities under different scenarios. Specifically, we test the versatility of ICR when considering different classifiers, alternating the algorithm used for optimization, and introducing shifts during the data generation. Also, all hyperparameter design choices we had to make are delineated in this part.

### 5.1 Reproducing Original Claims

Adopting the setup in the original study along with the released author's code König et al. (2022), we compare the performance of ICR against CE and CR in the synthetic and semi-synthetic dataset used in König et al. (2023). In alignment with the experimental description of the original study, we evaluate CE, individualized and subpopulation-based CR, and ICR for ten iterations, with each iteration consisting of five

model refits and four user-specified confidence levels for two hundred (200) individuals on each dataset. Table 6 in Appendix B contains the exact numerical values used for the reproducibility. We refer to these values as full-scale hyperparamters. König et al. (2023) used the outputs from all the experiments in order to answer four questions, relying on a different metric for each question; the observed improvement rate $\gamma_{obs}$ (Claim 1), the observed acceptance rates $\eta^{obs}$ (Claim 2), observed acceptance rates for other fits with comparable test set performance $\eta^{obs,refit}$ (Claim 3), and the average recourse cost for individuals who were rejected and were consequently provided with a recourse recommendation (Claim 4). Additionally, an invalidity metric is used for the robustness experiments, which expresses the percentage of post-recourse classifications that become invalid after the data has shifted (Rawal et al., 2020). More details on the metrics can be found in Appendix C.

## 5.2 Dataset Description

The authors have experimented with semi-synthetic and synthetic datasets in their study. The datasets comprise the SCMs and the corresponding directed acyclic graphs G. In addition to the four original datasets, we create an additional 3var-causal-nonlinear synthetic dataset. It is similar to the 3var-causal dataset used in the original experiments, but we introduce non-linearity through defining one of the features as a binomial distribution, and another one as a quadratic relation. The purpose of this new dataset is to compare the performance of CE, CR and ICR on a small, non-linear dataset that is lower in complexity compared to the semi-synthetic 5var-skill and 7var-covid datasets that the authors use. Table 1 showcases essential information for the datasets, while Appendix A provides more detailed information, as well as a visual depiction of the causal graphs and their structural equations.

| Name | Non-Linear | Features Affecting Y | Potential Gaming Variables (Features Affected by Y) | Source |
|---|---|---|---|---|
| 3var-causal | No | 3 | 0 | Synthetic |
| 3var-noncausal | No | 2 | 1 | Synthetic |
| 5var-skill | Yes | 2 | 3 | Semi-Synthetic |
| 7var-covid | Yes | 4 | 3 | Semi-Synthetic |
| 3var-causal-nonlinear | Yes | 3 | 0 | Synthetic |

Table 1: Information about the datasets

## 5.3 Robustness Assessment beyond original paper

While the original paper compares the robustness of CE, CR, and ICR on refits of the same data, we extend this robustness comparison to model and data shifts. This has been previously done on CE and CR (Upadhyay et al., 2021; Rawal et al., 2020) but, to the best of our knowledge, not on ICR. For this robustness comparison, we test different classifiers, shift the data, and use a different genetic algorithm. We had to down-scale the set of hyperparameters, to stay within our allocated ressources, by a factor of 2 for the 3var datasets. For the same reason, the experiments we conducted use only the datasets 3var-noncausal and 3var-causal and omitted the confidence values of 0.85 and 0.90. Table 2 summarizes the hyperparameters used in those experiments.

| Data set | Number of observations | Number of individuals having recourse calculation | Confidence | Number of Generations | POP SIZE | n digits | iterations |
|---|---|---|---|---|---|---|---|
| 3var-noncausal | 1000 | 100 | 0.75, 0.95 | 300 | 150 | 1 | 3 |
| 3var-causal | 1000 | 100 | 0.75, 0.95 | 300 | 150 | 1 | 3 |
| 3var-causal-nonlinear | 1000 | 100 | 0.75, 0.95 | 300 | 150 | 1 | 3 |

Table 2: Hyperparameters for the robustness experiments beyond the original paper.

### 5.3.1 Classifiers

The authors use random forest for classification, except in the *3var* datasets where logistic regression models are used. The former is utilized for non-linear datasets and the latter for linear ones. In this study, we compare the capabilities of ICR with different classifiers. The alternative classifiers tested are the AdaBoost Classifier (Schapire, 2013), a Support Vector Machine (SVM) for classification, and a simple Multi-Layer Perceptron (MLP). AdaBoost and SVM are implemented using the scikit-learn packages with the default hyperparameters (Pedregosa et al., 2011). The simple MLP consists of three hidden layers of 10, 10, and 5 nodes respectively. Adam is used for optimization (Kingma & Ba, 2014). ReLU is the activation function applied to all the layers.

### 5.3.2 Data shift

To create the data shift, we use the synthetic datasets 3var-causal and 3var-noncausal, where the features follow a standard normal distribution. We apply the same methodology as Upadhyay et al. (2021) and shift each dataset one feature at a time. Three settings can be distinguished: shifting the mean, the variance, and both simultaneously. The metric used is invalidity (Rawal et al., 2020). It measures the number of recourse recommendations that are not valid anymore for a model retrained on the shifted data. Details on this procedure can be found in Appendix D.

### 5.3.3 Genetic Algorithm

The authors employed a modified NSGA-II instance. This is done by altering the crowding distance computations, which are tailored for multi-objective counterfactual explanations as introduced in (Dandl et al., 2020) to minimize the cost of the optimization objective. In our study, we assess the capabilities of ICR by utilizing the newest version of Non-Dominated Sorting Genetic Algorithm (NSGA-III) (Deb & Jain, 2013). Building upon its predecessor NSGA-II, NSGA-III allows for improvements in diversity preservation and efficiency, promoting diversity among solutions. The optimization objective (described in Section 4.3) is a two-step problem modeled as a single-objective problem in the ICR original codebase. The capabilities of the two genetic algorithm variants in single-objective scenarios have not been widely studied as they are mainly used for multi-objective optimization. On top of that, the authors' implementation for NSGA-II is based on DEAP (Fortin et al., 2012)(evolutionary computation framework for rapid prototyping), which also supports NSGA-III natively. Thus, we firmly believe that comparing the two genetic algorithms is valuable for evaluating ICR for the given constrained optimization problem.

## 5.4 Computational requirements

The initial reproduction of the complete experiments proved quite computationally and time-intensive. After profiling the author's code, we found that using the original unoptimized code would not be viable for carrying out additional experiments for the robustness assessment of ICR. In order to make the experiment running process more cost-effective with respect to computation resources, the *multiprocess* python package is used to enable running the experiments for CE, CR, ICR, and the individualized/subpopulation settings simultaneously. Details on the speedup can be found in the Appendix H.

The experiments were carried out on a server using an AMD Rome CPU with 128 threads with a computational cost of 5-24 hours per single experiment setting while using parallelization and 10-55 hours without parallelization. We use our personal computers for the down-scaled experiments on an AMD Ryzen 5 5500U. In Appendix H, a visualization of the parallelization speedup captures the overall computational hours and a table documenting the CPU hours per dataset needed. The reproduction of the original results took an estimated 34 CPU hours after parallelization. It should be noted that the runtime would take up to around 100 CPU hours if it is run without the parallelization setting. The additional robustness and generalization experiments took an estimated 160 CPU hours in the parallelized setting while using the scaled-down version of the hyperparameters.

# 6 Results

In the upcoming subsections, we first compare our results with the authors' and validate which claims hold. We then further assess the robustness of ICR and compare its performance with CE and CR.

## 6.1 Results reproducing original paper

The results of our reproduction can be seen in Fig. 1. Fig. 1a shows the observed improvement rates $\gamma^{obs}$ for the different confidence intervals of CE, CR, and ICR. CE does not use confidence levels. Therefore, only one number is reported. Fig. 1b shows the observed acceptance rate $\eta^{obs}$. The robustness of refits from the same distribution can be seen in Fig. 1c. This graph shows the average acceptance rate of 5 refits. The average recourse cost can be seen in Fig. 1d. Appendix F is dedicated to a more detailed side-by-side comparison of the authors' outputs and ours.

**Claim 1:** It can be seen that only ICR has high improvement rates in Fig. 1a. CE and CR have very low improvement rates. The latter confirms the claim made by the authors. While the general trend still holds, the numbers retrieved during our experiments are very close to the ones provided by authors but not identical. Furthermore, it can be confirmed that CE and CR game on the 5var-skill dataset, by only applying recourse to the number of GitHub commits. In contrast, ICR suggests modifying values like years of experience and getting a degree, which are non-gaming variables. As a side effect of the causal model suggesting recourse actions on the years of experience and education, the number of commits also increases. Claim 1, which supports that ICR leads to improvement in situations where gaming is beneficial, can be confirmed.

**Claim 2:** CE, CR, and ICR all lead to high acceptance rates in Fig. 1b. Furthermore, it can be observed that ICR has lower acceptance levels than CE and CR. As such, we can confirm Claim 2. While we could not reproduce the exact numbers the authors provide, the general trends are the same. The subpopulation method performs worse than the individualized method.

**Claim 3:** The performance of the refitted models, which were created by sampling a new dataset from the SCM, varies per method as seen in Fig. 1c. CE and CR perform much worse on the refitted models, except on the 7var-covid dataset. This makes CE and CR not applicable to situations where the model will be refitted since the previous recourse recommendations could be invalidated, and the individual has to implement even more actions to achieve recourse. The acceptance rates of ICR are barely affected by refitting the models. This suggests that ICR is able to give recourse recommendations that will not change if a model is refitted with other data from the same distribution. This confirms Claim 3.

**Claim 4:** The recommendation costs, provided in Fig. 1d, are on average more expensive for ICR. This is due to the fact that ICR does not game by only applying the cheapest action, like the other methods do, often repeatedly. In the 5var-skill dataset ICR suggests getting a degree and gaining years of experience instead of creating a lot of commits, which makes it more costly than CE and CR. However, there are exceptions where ICR is cheaper than CE or CR, but on average, Claim 4 holds.

## 6.2 Results beyond original paper

**Classifiers:** We evaluate how robust a recourse recommendation is with respect to different classifiers. For the improvement rate, Table 3 indicates that the classifier does have an impact on the performance of CE and CR. The improvement rate of ICR is not dependent on the classifier, and therefore, we can argue that ICR is indeed more robust towards different decision algorithms. Notice that the reported $\gamma^{obs}$ values in Table 5 refer to the average improvement rate calculated across two synthetic datasets/SCMs (3var-causal & 3var-noncausal as in Table 1), using the reduced case hyper-parameters, as in Table 2. The acceptance performance is very similar regardless of the classifier being used. Using the refits for the acceptance, we provide some evidence that the classifier does not have a big impact on performance for CE, CR, and ICR; however, ICR has a slightly lower difference between the different classifiers. The detailed results analysis supports the latter observations on acceptance and acceptance under refits carried out in Appendix E. These results are important since in real life scenarios different types of models might be used to model a certain

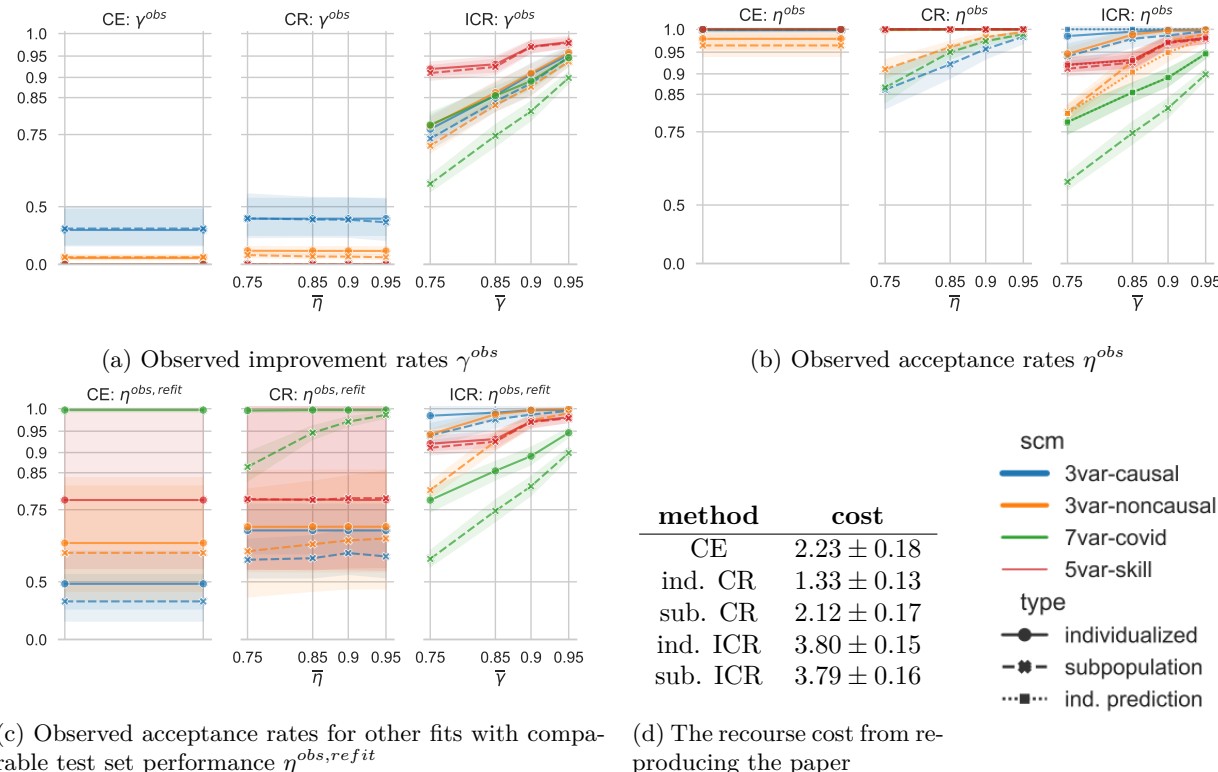

(a) Observed improvement rates $\gamma^{obs}$

(b) Observed acceptance rates $\eta^{obs}$

(c) Observed acceptance rates for other fits with comparable test set performance $\eta^{obs,refit}$

(d) The recourse cost from reproducing the paper

| method | cost |
|---|---|
| CE | $2.23 \pm 0.18$ |
| ind. CR | $1.33 \pm 0.13$ |
| sub. CR | $2.12 \pm 0.17$ |
| ind. ICR | $3.80 \pm 0.15$ |
| sub. ICR | $3.79 \pm 0.16$ |

Figure 1: Experimental results for CE, CR and ICR for the reproducibility

task and it can be seen that CE and CR show different performances depending on the model, while ICR performs well independently of the model. ICR is therefore be better applicable to real life scenarios.

| recourse | MLP | SVM | adaboost | logreg | random forest |
|---|---|---|---|---|---|
| CE | $0.29 \pm 0.11$ | $0.26 \pm 0.1$ | $0.19 \pm 0.04$ | $0.28 \pm 0.15$ | $0.26 \pm 0.02$ |
| ind. CR | $0.33 \pm 0.08$ | $0.32 \pm 0.10$ | $0.20 \pm 0.01$ | $0.31 \pm 0.16$ | $0.27 \pm 0.03$ |
| ind. ICR | $0.95 \pm 0.00$ | $0.95 \pm 0.00$ | $0.95 \pm 0.01$ | $0.96 \pm 0.00$ | $0.95 \pm 0.01$ |
| sub. CR | $0.31 \pm 0.13$ | $0.31 \pm 0.13$ | $0.18 \pm 0.02$ | $0.31 \pm 0.16$ | $0.24 \pm 0.04$ |
| sub. ICR | $0.95 \pm 0.00$ | $0.95 \pm 0.00$ | $0.95 \pm 0.01$ | $0.95 \pm 0.01$ | $0.96 \pm 0.00$ |

Table 3: $\gamma^{obs}$ for different classifiers with specified confidence of 0.95, on the reduced case scenario

**Data shift:** To further assess the robustness of our different recourse methods, the data is shifted and it is compared whether a refit of the model leads to invalidation of previous recommendations. Since CE and CR already struggle with refits from the same distribution, it is of no surprise that CE and CR perform even worse when the distribution slightly changes. Variance shifts seem to be slightly worse for the model invalidity than mean shifts. ICR, on the other hand, does not show any of these issues. As can be seen from Table 4, the highest amount of invalidity appears in the subpopulation approach with 6%. This means 6% of the previous recourse recommendations are not valid anymore after the model was refitted on the shifted data. This suggests that ICR is not only robust to refits from the same distribution but also to refits from distribution shifts to a certain extent. A method that has low invalidity to data shifts, is better applicable in real life scenarios as the distribution in the data can change over time and a low invalidity means that the model can handle the old data as well as the shifted data.

A more detailed comparison of distributional changes and models would be optimal here; however, due to

the computational resources necessary to calculate recourse, only a small sample of data shifts and models are compared here. All values in Table 4 show the average invalidity calculated across 3var-causal and 3var-noncausal datasets/SCMs.

| recourse | classifier | both shift | variance shift | mean shift |
|---|---|---|---|---|
| CE | MLP | $0.34 \pm 0.34$ | $0.86 \pm 0.23$ | $0.82 \pm 0.34$ |
| | SVM | $0.35 \pm 0.36$ | $0.89 \pm 0.22$ | $0.85 \pm 0.32$ |
| | adaboost | $0.77 \pm 0.11$ | $0.90 \pm 0.06$ | $0.89 \pm 0.06$ |
| | logreg | $0.46 \pm 0.34$ | $0.92 \pm 0.16$ | $0.84 \pm 0.33$ |
| | rf | $0.78 \pm 0.08$ | $0.90 \pm 0.07$ | $0.91 \pm 0.05$ |
| ind. CR | MLP | $0.30 \pm 0.30$ | $0.84 \pm 0.21$ | $0.80 \pm 0.31$ |
| | SVM | $0.32 \pm 0.31$ | $0.88 \pm 0.20$ | $0.84 \pm 0.28$ |
| | adaboost | $0.73 \pm 0.11$ | $0.88 \pm 0.08$ | $0.87 \pm 0.06$ |
| | logreg | $0.40 \pm 0.27$ | $0.90 \pm 0.15$ | $0.83 \pm 0.30$ |
| | rf | $0.75 \pm 0.07$ | $0.88 \pm 0.08$ | $0.88 \pm 0.07$ |
| ind. ICR | MLP | $0.05 \pm 0.13$ | $0.02 \pm 0.03$ | $0.01 \pm 0.02$ |
| | SVM | $0.04 \pm 0.10$ | $0.00 \pm 0.01$ | $0.00 \pm 0.01$ |
| | adaboost | $0.05 \pm 0.09$ | $0.04 \pm 0.04$ | $0.04 \pm 0.04$ |
| | logreg | $0.04 \pm 0.10$ | $0.01 \pm 0.02$ | $0.00 \pm 0.01$ |
| | rf | $0.04 \pm 0.07$ | $0.05 \pm 0.09$ | $0.05 \pm 0.08$ |
| sub. CR | MLP | $0.33 \pm 0.32$ | $0.84 \pm 0.22$ | $0.81 \pm 0.34$ |
| | SVM | $0.34 \pm 0.32$ | $0.87 \pm 0.22$ | $0.84 \pm 0.31$ |
| | adaboost | $0.77 \pm 0.11$ | $0.90 \pm 0.05$ | $0.90 \pm 0.07$ |
| | logreg | $0.41 \pm 0.31$ | $0.90 \pm 0.15$ | $0.82 \pm 0.32$ |
| | rf | $0.79 \pm 0.07$ | $0.92 \pm 0.06$ | $0.90 \pm 0.06$ |
| sub. ICR | MLP | $0.02 \pm 0.06$ | $0.03 \pm 0.04$ | $0.02 \pm 0.02$ |
| | SVM | $0.02 \pm 0.06$ | $0.01 \pm 0.02$ | $0.01 \pm 0.02$ |
| | adaboost | $0.05 \pm 0.05$ | $0.05 \pm 0.06$ | $0.06 \pm 0.05$ |
| | logreg | $0.02 \pm 0.06$ | $0.02 \pm 0.03$ | $0.01 \pm 0.01$ |
| | rf | $0.03 \pm 0.05$ | $0.05 \pm 0.08$ | $0.06 \pm 0.09$ |

Table 4: Invalidity for the shifted features, averaged across 3var-causal and 3var-noncausal. The demonstrated results are the average and standard deviation for shifts overall features and three iterations, with a user-specified confidence level set to 0.95.

**Genetic algorithms:** Table 5 compares the yielded improvement rates $\gamma^{obs}$ between the modified NSGA-II variant, utilized in (König et al., 2023) and NSGA-III (Deb & Jain, 2013) for minimizing the optimization objective in our disposal, as defined in Subsection 4.3. The figures present in Table 5 refer to the average improvement rate calculated across two synthetic datasets/SCMs (3var-causal & 3var-noncausal as in Table 1), using the reduced case hyper-parameters. Both genetic algorithms acquire similar performance when targeting improvement. Additional experiments for acceptance rate $\eta^{obs}$ and $\eta^{obs,refit}$ further compare the two different algorithms and are provided in Appendix G, Tables 9 and 10 respectively. Interestingly, when considering the acceptance under refits, NSGA-III in most cases yield similar, if not higher rates.

**New Dataset:** To further assess the generalizability of ICR, we test its performance on a synthetic dataset we created and refer to it as the 3var-causal-nonlinear[1]. The results for the observed improvement $\gamma^{obs}$, acceptance rates $\eta^{obs}$ are shown in Fig. 2a and Fig. 2b, respectively. Additionally, Fig.2c depicts the robustness of refits from the same distribution.

CR attains a perfect rate for acceptance, while individualized ICR and CE follow closely behind. As expected, CE and CR obtain low improvement rate values, whereas ICR consistently leads to improvement for both user-specified confidence levels. Ultimately, ICR seems to be the prevailing method when testing for robustness regarding refits on the same data, and supports the empirical claims made by the authors.

---

[1]Avid readers can find the SCM model along with the structural equations in the AppendixA

| recourse | NSGA-II | NSGA-III |
|:---:|:---:|:---:|
| CE | $0.28 \pm 0.13$ | $0.31 \pm 0.13$ |
| ind. CR | $0.32 \pm 0.14$ | $0.33 \pm 0.12$ |
| ind. ICR | $0.96 \pm 0.01$ | $0.98 \pm 0.02$ |
| sub. CR | $0.31 \pm 0.15$ | $0.32 \pm 0.14$ |
| sub. ICR | $0.95 \pm 0.03$ | $0.95 \pm 0.02$ |

Table 5: $\gamma^{obs}$ (observed rate $\pm$ standard deviation) of each genetic algorithm achieved with user-specified confidence of 0.95 on the reduced hyper-parameter case scenario. All rates in the table have been rounded to the third decimal place.

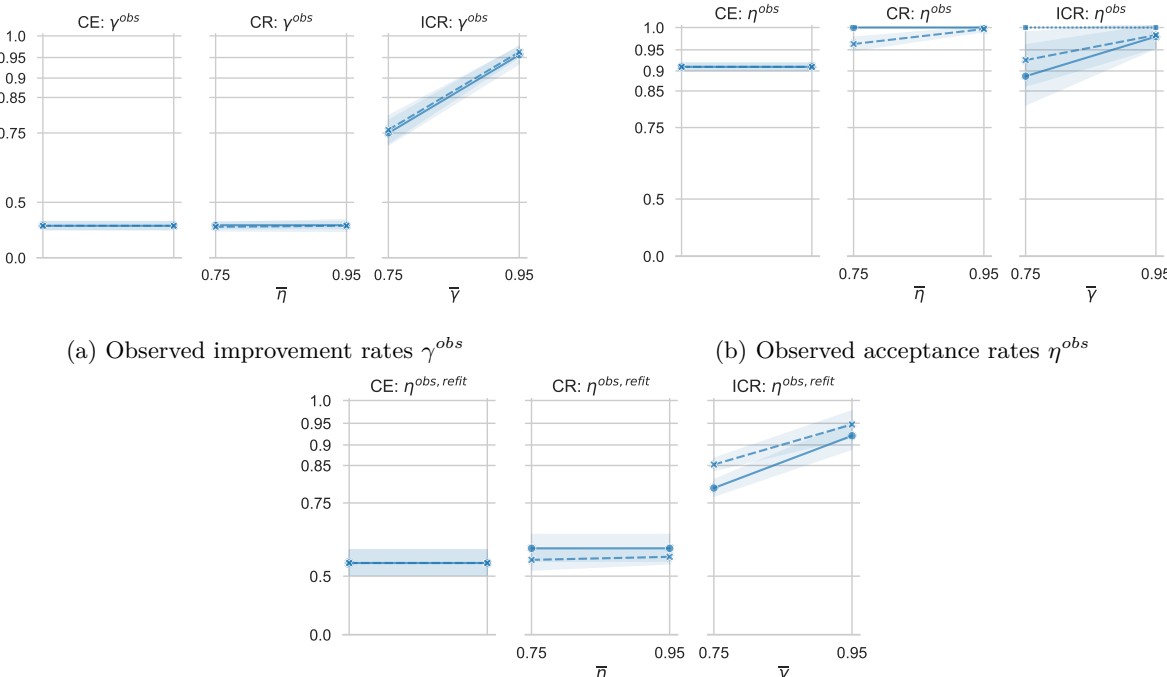

(a) Observed improvement rates $\gamma^{obs}$

(b) Observed acceptance rates $\eta^{obs}$

(c) Observed acceptance rates for other fits with comparable test set performance $\eta^{obs,refit}$

Figure 2: Experimental results for CE, CR and ICR on the 3var-causal-nonlinear dataset

# 7 Challenges during Reproducibility

Overall, the public repository containing the original code was well-structured and documented. The provided scripts to produce and visualize the results were very helpful, and thus, analyzing and comparing our results was reasonably straightforward. Furthermore, the original paper provided detailed information on implementation details and theoretical background, which is presented in the study's appendix.

However, inconsistencies exist between the hyperparameter value choices in the experiment details specified in the paper and those in the repository instructions. Specifically, for the 7var-covid dataset, a lower value is selected for the number of generations and population size in the repository compared to the description in the paper. Additionally, the number of runs performed per dataset hyperparameter differs in the plots presented in the original paper and the author's repository description. Our hyperparameter selection follows the paper specifications as delineated in Section 5.

We discovered that the original author's implementation resulted in different numerical results when repeating the same experiment, originating from a seeding issue. Indeed, in the provided code in (König et al., 2022) the values for each SCM of the datasets are sampled randomly from corresponding distributions each time, making this generation process non-deterministic. We adjust our implementation accordingly to handle this issue, and all reported results are retrieved by setting the seed to 1. Finally, the requirements list provided by the authors for installing the packages needed for the project did not work out of the box, and we had some package dependency issues. These dependencies were resolved by reverting some of the packages back to the relevant versions[2] when the original paper was published.

**Communication with the authors** We have contacted the paper's first author to ask for clarification of the theoretical aspects and some technical parts of the code. Moreover, we asked for some feedback on the proposed extensions. After 2 weeks the author responded to all of our questions, while he found our extensions interesting and provided constructive feedback on them. A new direction for research was also proposed, not implemented in the author's paper but was only hinted at. Regarding the discrepancies discussed here the corresponding author recognized what we have pointed out and sub-sequentially updated the repository description accordingly.

# 8 Discussion

König et al. (2023) distinguish two purposes for contrastive explanations: contestability of algorithmic decisions and actionable recourse recommendations. ICR targets improvement, which is a necessity of actual recourse. Thus, recourse is achieved by improving the underlying condition rather than just the features that can game the predictor model. The most significant limitation of ICR is that a causal graph or an SCM is needed. These are not always available, thus limiting the applicability of ICR.

Our contributions were twofold: firstly, we reproduced the experiments by König et al. (2023) and provided evidence of their claims' validity. While it was impossible to replicate the exact numbers of the authors due to how the seeds were set, we can replicate the trends of all claims. Secondly, we assessed the robustness of the ICR claim against different model fits and data fits, as well as the generalizability across a new dataset. Our additional experiments tested different classifiers, an alternative genetic algorithm for minimizing the optimization objective, and the robustness to mean and variance shifts in the dataset.

To test the influence of genetic algorithms in the minimization objective, we adapted the NSGA-III algorithm to also make use of the same crowding distance and principles as in the NSGA-II variant used by the authors, inspired from (Dandl et al., 2020), we discovered that it performs equally well and even attains better outcome at some dataset/SCMs runs. Since these very similar genetic algorithms acquire similar performance, it would be interesting to try out other evolutionary and/or genetic algorithms that specifically target single-objective functions, aiming to derive better recourse recommendations and reduce the computation time spent during the optimization phase. One research direction that has yet to be explored is effectively using the multi-objective capabilities of the two NSGA variants. As for future research, we aim to optimize for improvement

---

[2]In our repository, we provide the update .yml files for anyone interested in working with our work.

and acceptance rate jointly, which was suggested during our correspondence with the author. Specifically, one could jointly target improvement rate and cost and let the user choose from the Pareto front.

Concerning the robustness assessment, we can verify to a greater degree that ICR is more robust than CE and CR, specifically towards mean and variance shifts in the data. However, given more computational resources, we would like to conduct a more extensive assessment by testing different magnitudes in shifts. As for the generalizability experiment with the additional dataset 3var-causal-nonlinear, the evidence partially points towards the generalizability strength of ICR since the trends are similar to the performance for the larger non-linear datasets. Nevertheless, they also closely follow the performance trends of the other 3var datasets.

A possible limitation of our experiments on robustness is that we ran them on a down-scaled set of hyperparameters. Even though running on the complete set of hyperparameters would make a difference in the reliability of our conclusions, we must recognize the significant computational resources that the original experiments require. The environmental impact of computationally expensive methods is a solid motivation for further research into making ICR more efficient and effective.

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

# A   Dataset Information

This section will provide more information about the dataset we used. Moreover, we have added each dataset's causal graph and structural equation in the following figures 3 - 7.

**3var-causal:** A linear Gaussian SCM with a binary target Y, having all other features influencing it.

**3var-noncausal:** Similar to the 3var-causal, but one feature is affected by Y.

**5var-skill:** A categorical semi-synthetic SCM where the target is the programming skill level based on causes like university degree and non-causal factors obtained from GitHub such as commit count. This dataset was inspired by Montandon et al. (2021)

**7var-covid:** A semi-synthetic dataset replicated by a real-world COVID screening model provided by (Jehi et al., 2020). The model has causes like COVID-19 vaccination and population density, including symptoms like fever and fatigue. The dataset illustrates a mix of categorical and continuous data with various noise distributions. Their relationships include nonlinear structural equations.

**3var-causal-nonlinear** A fully synthetic non-linear SCM with a binary target Y, having all other features influencing it.

In the following cost equations, we define $\delta$ as the vector of absolute changes to the intervened-upon variables.

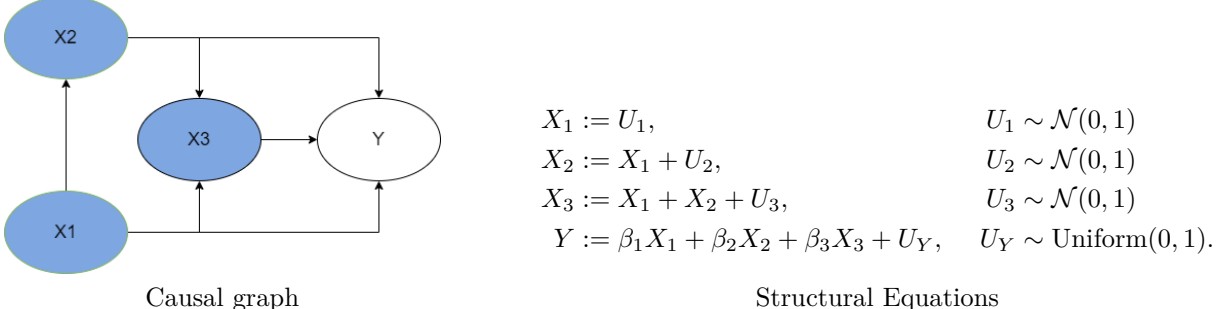

$$
\begin{aligned}
X_1 &:= U_1, & U_1 &\sim \mathcal{N}(0,1) \\
X_2 &:= X_1 + U_2, & U_2 &\sim \mathcal{N}(0,1) \\
X_3 &:= X_1 + X_2 + U_3, & U_3 &\sim \mathcal{N}(0,1) \\
Y &:= \beta_1 X_1 + \beta_2 X_2 + \beta_3 X_3 + U_Y, & U_Y &\sim \mathrm{Uniform}(0,1).
\end{aligned}
$$

Causal graph    Structural Equations

Figure 3: SCM for 3var-causal with $\mathrm{cost}(a) = \delta_1 + \delta_2 + \delta_3$

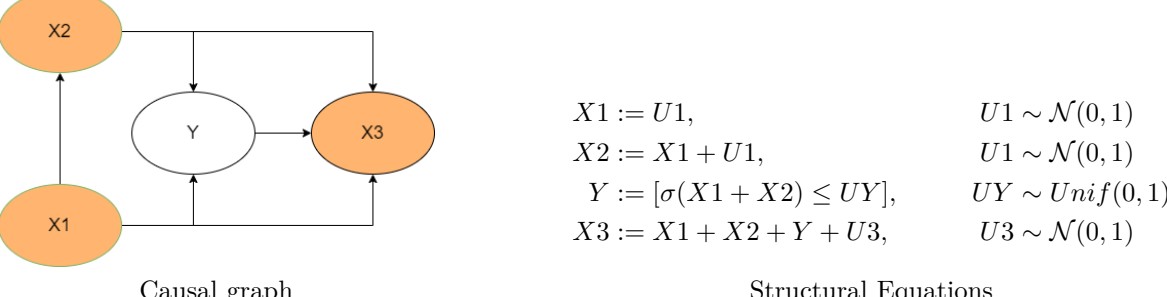

$$
\begin{aligned}
X1 &:= U1, & U1 &\sim \mathcal{N}(0,1) \\
X2 &:= X1 + U1, & U1 &\sim \mathcal{N}(0,1) \\
Y &:= [\sigma(X1 + X2) \leq UY], & UY &\sim Unif(0,1) \\
X3 &:= X1 + X2 + Y + U3, & U3 &\sim \mathcal{N}(0,1)
\end{aligned}
$$

Causal graph    Structural Equations

Figure 4: SCM for 3var-noncausal with $\mathrm{cost}(a) = \delta_1 + \delta_2 + \delta_3$

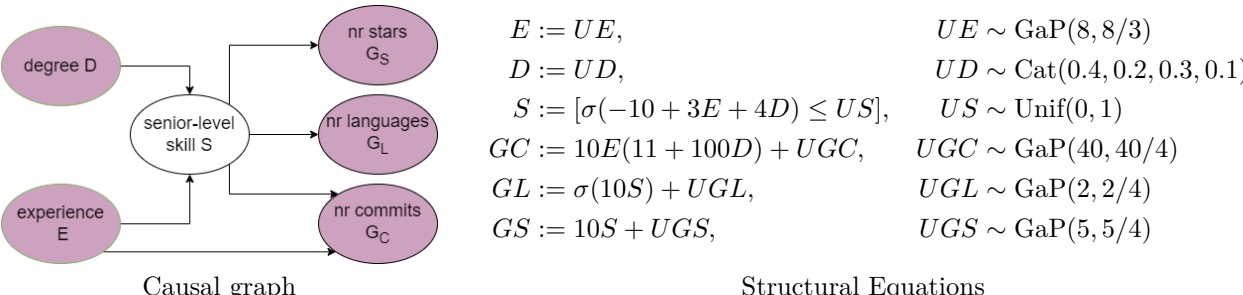

$$E := UE, \qquad\qquad UE \sim \mathrm{GaP}(8, 8/3)$$
$$D := UD, \qquad\qquad UD \sim \mathrm{Cat}(0.4, 0.2, 0.3, 0.1)$$
$$S := [\sigma(-10 + 3E + 4D) \le US], \quad US \sim \mathrm{Unif}(0,1)$$
$$GC := 10E(11 + 100D) + UGC, \quad UGC \sim \mathrm{GaP}(40, 40/4)$$
$$GL := \sigma(10S) + UGL, \qquad\qquad UGL \sim \mathrm{GaP}(2, 2/4)$$
$$GS := 10S + UGS, \qquad\qquad UGS \sim \mathrm{GaP}(5, 5/4)$$

Causal graph       Structural Equations

Figure 5: SCM for 5var-skill with cost(a) $= = 5\delta_E + 5\delta_D + 0.0001\delta_{G_C} + 0.01\delta_{G_L} + 0.1\delta_{G_S}$

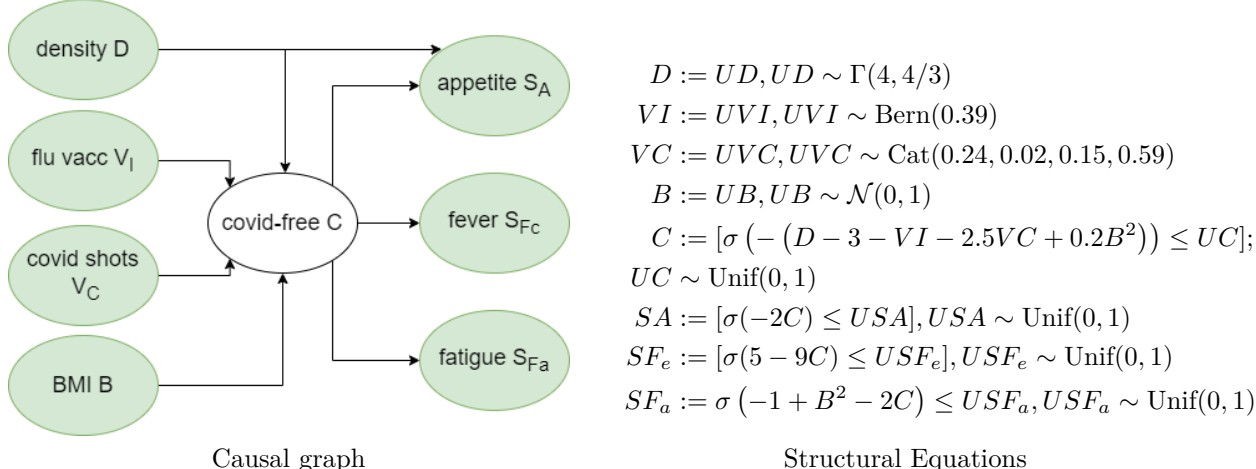

$$D := UD, UD \sim \Gamma(4, 4/3)$$
$$VI := UVI, UVI \sim \mathrm{Bern}(0.39)$$
$$VC := UVC, UVC \sim \mathrm{Cat}(0.24, 0.02, 0.15, 0.59)$$
$$B := UB, UB \sim \mathcal{N}(0,1)$$
$$C := [\sigma\left(-\left(D - 3 - VI - 2.5VC + 0.2B^2\right)\right) \le UC];$$
$$UC \sim \mathrm{Unif}(0,1)$$
$$SA := [\sigma(-2C) \le USA], USA \sim \mathrm{Unif}(0,1)$$
$$SF_e := [\sigma(5 - 9C) \le USF_e], USF_e \sim \mathrm{Unif}(0,1)$$
$$SF_a := \sigma\left(-1 + B^2 - 2C\right) \le USF_a, USF_a \sim \mathrm{Unif}(0,1)$$

Causal graph       Structural Equations

Figure 6: SCM for 7var-covid with cost(a) $= \delta_D + \delta_{VI} + \delta_{VC} + \delta_B + \delta_{SA} + \delta_{SF_e} + \delta_{SF_a}$.

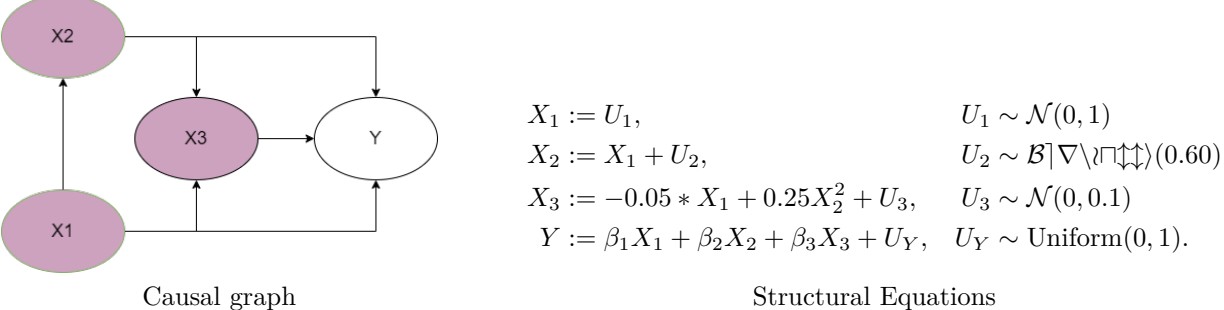

$$X_1 := U_1, \qquad\qquad U_1 \sim \mathcal{N}(0,1)$$
$$X_2 := X_1 + U_2, \qquad\qquad U_2 \sim \mathcal{B}|\nabla \backslash \wr \sqcap \updownarrow \rangle (0.60)$$
$$X_3 := -0.05 * X_1 + 0.25X_2^2 + U_3, \quad U_3 \sim \mathcal{N}(0, 0.1)$$
$$Y := \beta_1 X_1 + \beta_2 X_2 + \beta_3 X_3 + U_Y, \quad U_Y \sim \mathrm{Uniform}(0,1).$$

Causal graph       Structural Equations

Figure 7: SCM for 3var-nonlinear with cost(a) $= \delta_1 + \delta_2 + \delta_3$

## B   Hyperparameters for reproducibility study

Table 6 presents the hyperparameters used to reproduce the author's results. The number of observation column refer to the dataset, whereas the population size and the number of generations columns are relevant to the genetic algorithm optimization procedure. The confidence hyperparametrer has different interpretation depending which model we are evaluating. Since Counterfactual Explanations (CE) prediction function is deterministic, meaning that changing the inputs to the model such that the prediction change in the desired way. Thus the confidence hyperparameter does not apply in the CE method. Contrastively, this hyperparamter defines the targeted acceptance probability and the target improvement probability for the CR and ICR respectively.

| Data set | Number of observations | Number of individuals having recourse calculation | Confidence | Number of Generations | POP SIZE | n digits | nr refits |
|---|---|---|---|---|---|---|---|
| 3var-noncausal | 4000 | 200 | 0.75, 0.85, 0.9, 0.95 | 600 | 300 | 1 | 5 |
| 3var-causal | 4000 | 200 | 0.75, 0.85, 0.9, 0.95 | 600 | 300 | 1 | 5 |
| 5var-skill | 4000 | 200 | 0.75, 0.85, 0.9, 0.95 | 1000 | 500 | 1 | 5 |
| 7var-covid | 20000 | 200 | 0.75, 0.85, 0.9, 0.95 | 1000 | 500 | 1 | 5 |

Table 6: Hyperparameters based on the original paper.

## C   Experiment metrics

**Experiment 1:** Do CE, CR, and ICR lead to improvement? The observed improvement rates $\gamma_{obs}$ was the metric to assess the data. In the setting where the structural equations are assumed, it is possible to acquire individualized improvement confidence. The subpopulation-based improvement confidence is derived in a setting where only the causal graph is assumed.

**Experiment 2:** Do CE, CR, and ICR lead to acceptance (by pre- and post-recourse predictor)? Recourse recommendations should lead to improvement and change the classifier's original decision. Whether acceptance naturally ensues from the improvement rate depends on the ability of the predictor to recognize improvements. Thus, the metric calculated here is the observed acceptance rates $\eta^{obs}$ w.r.t. the optimal pre-recourse observational predictor $h^*$; and in the case of individualized ICR additionally w.r.t. the individualized post-recourse predictor $h^*_{ind}$, in order to account for an imbalance between ICR and the predictor.

**Experiment 3:** Do CE, CR, and ICR lead to acceptance by other predictors with comparable test error? The metric deployed for the experiment is the observed acceptance rates for other fits with comparable test set performance $\eta^{obs,refit}$

**Experiment 4:** How costly are CE, CR, and ICR recommendations? For the last experiment, the authors used the average recourse cost for rejected individuals and were consequently provided with a recourse recommendation. The cost is defined differently for each dataset and can be found in Appendix 3 of the original paper.

## D   Robustness on shifted data

The 3var datasets consist of Standard Normal Distributions (mean 0 and variance 1) that are causally related. For each feature, we create a new dataset by shifting once the mean (from 0 to 0.5), once the variance (from 1.0 to 0.5), and once both (mean from 0 to 0.5 and variance from 1.0 to 0.5). Due to the causal relationships, a shift for x1 also affects all children of x1. A similar procedure for shifting the data was used by Upadhyay et al. (2021).
We have our unshifted data $D_1$ and model $M_1$, which is trained on $D_1$. Now, we shift one feature by a specific mean or variance or both and then create a new dataset $D_2$. On this data, we create the model

$M_2$. We used 50% for model training and 50% for the validation, like König et al. (2023) did. Recourse is applied to data $D_1$ to revert the decision of $M_1$. Invalidity calculates how many individuals' recourse recommendations are invalid after the data shift. To implement this, the recourse recommendation is used as an input for $M_2$, and a recourse is marked as invalid if $M_2$ predicts the decision as 0. Therefore, the recourse recommendation did not change the algorithm's decision. The process of calculating the invalidity was implemented by Rawal et al. (2020).

## E  Robustness of the classifier

Tables 7 and 8 depict our experimental results for the robustness of ICR when considering different classifiers.

| recourse | MLP | SVM | adaboost | logreg | random forest |
|---|---|---|---|---|---|
| CE | $0.98 \pm 0.00$ | $0.98 \pm 0.00$ | $0.93 \pm 0.03$ | $0.96 \pm 0.05$ | $0.88 \pm 0.04$ |
| ind. CR | $1.00 \pm 0.00$ | $1.00 \pm 0.00$ | $1.00 \pm 0.00$ | $1.00 \pm 0.00$ | $1.00 \pm 0.00$ |
| ind. ICR | $1.00 \pm 0.01$ | $1.00 \pm 0.00$ | $0.98 \pm 0.01$ | $1.00 \pm 0.00$ | $0.99 \pm 0.00$ |
| sub. CR | $0.99 \pm 0.00$ | $0.98 \pm 0.02$ | $1.00 \pm 0.00$ | $0.99 \pm 0.00$ | $1.00 \pm 0.01$ |
| sub. ICR | $1.00 \pm 0.01$ | $1.00 \pm 0.00$ | $0.98 \pm 0.02$ | $1.00 \pm 0.00$ | $0.99 \pm 0.01$ |

Table 7: $\eta^{obs}$ of different classifiers with confidence 0.95, reduced datasets for the classifiers

| recourse | MLP | SVM | adaboost | logreg | random forest |
|---|---|---|---|---|---|
| CE | $0.42 \pm 0.15$ | $0.41 \pm 0.12$ | $0.39 \pm 0.01$ | $0.54 \pm 0.1$ | $0.42 \pm 0.01$ |
| ind. CR | $0.48 \pm 0.13$ | $0.46 \pm 0.11$ | $0.41 \pm 0.01$ | $0.59 \pm 0.08$ | $0.45 \pm 0.01$ |
| ind. ICR | $1.00 \pm 0.01$ | $1.00 \pm 0.00$ | $0.98 \pm 0.02$ | $1.00 \pm 0.0$ | $0.98 \pm 0.02$ |
| sub. CR | $0.44 \pm 0.17$ | $0.43 \pm 0.13$ | $0.41 \pm 0.0$ | $0.57 \pm 0.09$ | $0.43 \pm 0.01$ |
| sub. ICR | $0.99 \pm 0.01$ | $1.00 \pm 0.00$ | $0.98 \pm 0.02$ | $1.00 \pm 0.00$ | $0.97 \pm 0.02$ |

Table 8: $\eta^{obs,refit}$ of different classifiers with confidence 0.95, reduced datasets for the classifiers

## F  Trend comparison

In this section, we provide all the results of König et al. (2023) next to our findings for the reproducibility part.

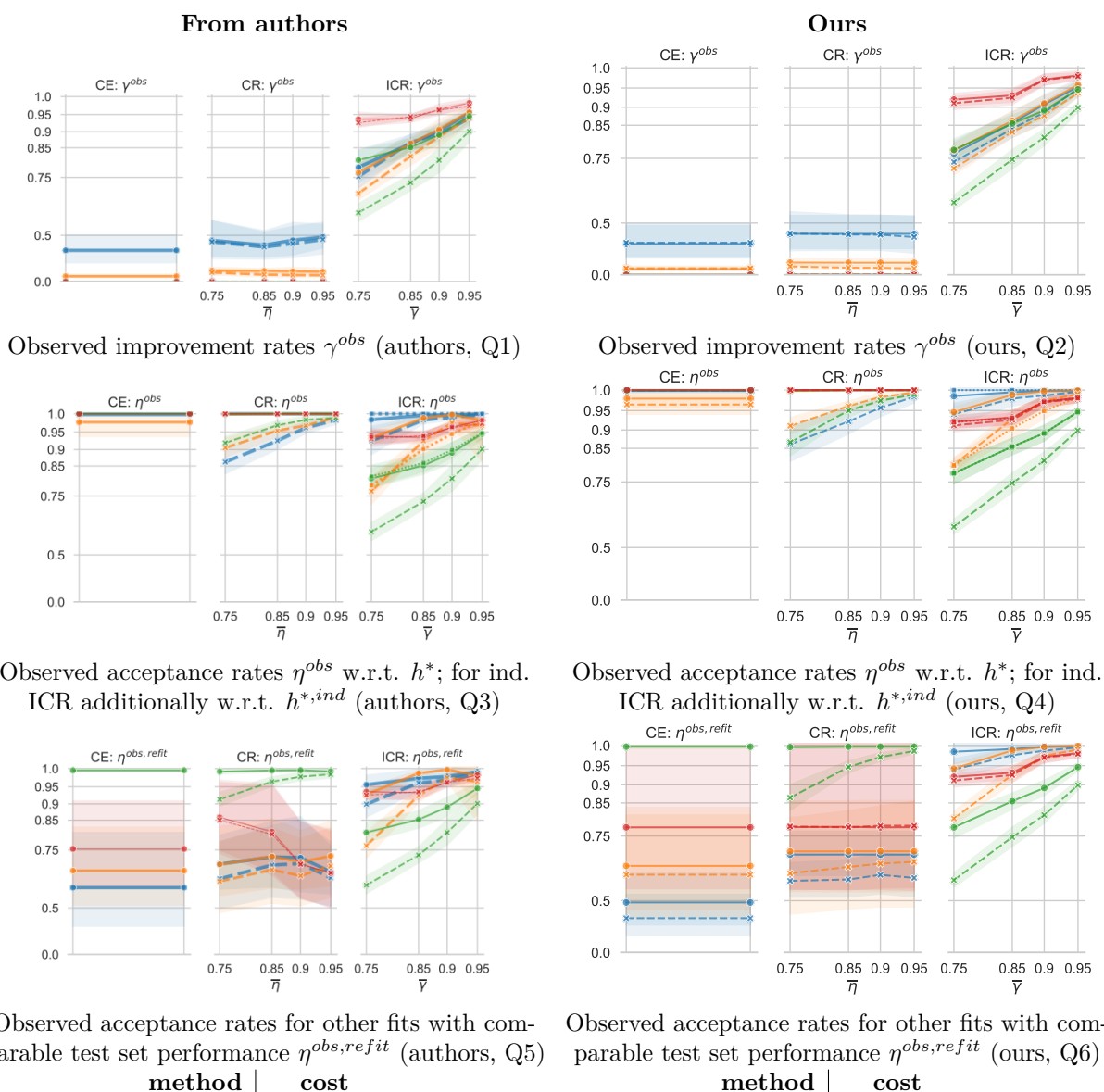

Figure 8: Trend comparison

# G    Robustness of the genetic algorithm

The following Tables 9 and 10 depict the experimental results for acceptance rate $\eta^{obs}$ and $\eta^{obs,refit}$, respectively, to compare the robustness of ICR when using two different algorithms for minimizing the specified optimization problem.

| recourse | NSGA-II | NSGA-III |
|:---:|:---:|:---:|
| CE | $0.96 \pm 0.07$ | $0.96 \pm 0.08$ |
| ind. CR | $1.00 \pm 0.00$ | $1.00 \pm 0.00$ |
| ind. ICR | $1.00 \pm 0.00$ | $0.99 \pm 0.00$ |
| sub. CR | $0.99 \pm 0.01$ | $0.99 \pm 0.01$ |
| sub. ICR | $0.99 \pm 0.00$ | $0.99 \pm 0.00$ |

Table 9: $\eta^{obs}$ (observed rate $\pm$ standard deviation) of each genetic algorithm achieved for user-specified confidence of 0.95, using the reduced case scenario hyper-parameters, as in Table 2. All rates present in the table have been rounded to the third decimal place.

| recourse | NSGA-II | NSGA-III |
|:---:|:---:|:---:|
| CE | $0.54 \pm 0.16$ | $0.54 \pm 0.12$ |
| ind. CR | $0.59 \pm 0.13$ | $0.59 \pm 0.09$ |
| ind. ICR | $1.00 \pm 0.00$ | $0.99 \pm 0.00$ |
| sub. CR | $0.57 \pm 0.15$ | $0.57 \pm 0.12$ |
| sub. ICR | $0.99 \pm 0.00$ | $0.99 \pm 0.00$ |

Table 10: $\eta^{obs,refit}$ (observed rate $\pm$ standard deviation) of each genetic algorithm achieved for user-specified confidence of 0.95, using the reduced case scenario hyper-parameters, as in Table 2. All rates present in the table have been rounded to the third decimal place.

## H    Computational Resources

In the following Figure 9, the effect of multiprocessing on the reproduction speed of different methods is illustrated.

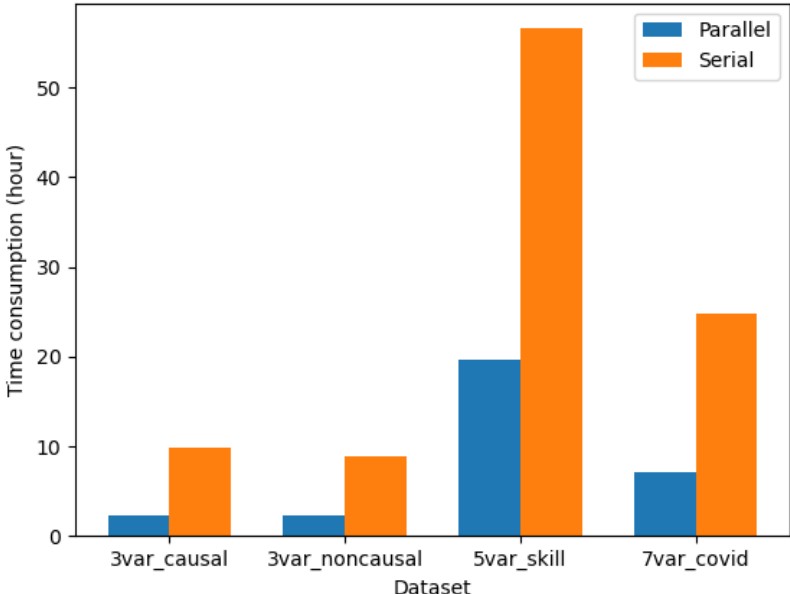

Figure 9: Time usage comparison for the original experiments with original (linear) method and the improved one (parallel)

