# OpenReview forum: "On the Reproducibility of: Improvement-Focused Causal Recourse"
_TMLR — Rejected by TMLR_

### Review · Reviewer_F393 · 2024-03-22

**Summary Of Contributions:**

The authors reproduce the main empirical findings from the paper "Improvement-Focused Causal Recourse (ICR)" and extend the experiments to additionally assess the robustness of ICR and competitors with respect to distribution shifts. Moreover, they extend the experiments to account for more model classes, a newer version of the NSGA optimizer, and one additional dataset. They find that the claims from the original paper can be reproduced; moreover, the superior robustness of ICR is demonstrated to extend to distribution shifts in an illustrative example (in contrast to just model refits).

**Audience:**

Yes

**Broader Impact Concerns:**

There is no broader impact statement. A discussion of the results' implications, especially regarding the robustness, would improve the paper. However, I don't think it is absolutely necessary to add one.

**Claims And Evidence:**

Yes

**Requested Changes:**

- p1, "may suggest actions that revert the model's verdict whenever possible": More correct would be "whenever lucrative".
- p1, "shifts away from the counterfactual explanations paradigm": A bit unclear what is meant with this paradigm. Maybe write something like "Karimi et al recognize CR to be a causal problem" instead?
- p1, "to design decision systems that accurately predict both pre- and post-recourse." Not quite correct; Changing the decision system is only a small part of the ICR framework and only applies in the context of individualized recourse (where the SCM is known). The causal knowledge is mostly leveraged to make improvement-focused recommendations (and especially causal graphs are used for nothing else). The follow up sentence makes no sense; do you really mean the model adaption when saying "this is done by defining the improvement confidence"?
- p2, "are one of the human-friendly approaches", can you support this with a citation?
- p2, "improves on this by considering the causal dependencies between covariates": That was the contribution of CR, the contribution of ICR is to consider the causal dependencies with Y.
- p2, "there is a minor inconsistency": There would be an inconsistency if different values were reported in the paper (which is not the case). Maybe rephrase along the lines of: "For the number of observations, we used the specification from the repo since not specified in the paper."
- p2, "SCM recourse-based" -> "SCM-based recourse"
- p3, "knowledge of the SCM is unknown" -> "where the SCM is not known"
- p3, "no observed variables influence both the dependent and independent variables" -> this is not generally true. It is still an assumption.
- p4, "in order to conduct additional" -> I was confused for a second where the changes to the hyperparameters apply. Maybe start the paragraph with a sentence alone the lines of "We additionally performed experiments regarding the robustness of the methods. For the robustness experiments, we down-scaled ..."
- p4, Overall, the text in Section 3 mixes describing background, describing what you did, describing what is novel/different, and discussing challenges when reproducing the results. Especially for the described problems in reproduction, it is difficult to get an overview, and difficult to understand how/whether the inconsistencies could be resolved in conversation with the authors. I would suggest restructuring Section 3, for example, as follows:
   - move the background about ICR to a background section, and also explain CE, and CR
   - In Section 3 focus on clearly delivering which experiments you reproduced and which additional experiments you conducted.
  - Add a separate Section to Section 3 describing and discussing in one place what challenges you faced when reproducing the results (e.g., hyperparameter only mentioned in the repo), how you resolved those challenges, what the authors had to say about it, etc.
- p6, "games by only applying recourse to the number of commits" -> This only applies to the skill dataset, which is unclear from your text.
- p8, "This lead to the conclusion that ICR is robust not only to refits from the same distribution but even to refits from distributions shifts" -> The conclusion is a bit strong, given that only a few small synthetic examples were studied. Maybe reword to something like "suggests that ICR is robust ...".
- p10, "and the [robustness to ] mean and variance shifts in the dataset."
- p10 "Although a bit late" - What does that mean? Two weeks or two months?

**Strengths And Weaknesses:**

Strengths:
- The authors carefully reproduced the results of a recent and relevant paper. Knowing that the comparison between the three methods regarding claims 1-4 can be reproduced will interest authors in the field.
- The extension of the results to more types of classifiers -- especially when it comes to the robustness of recourse to refits -- is interesting, and supports the theoretical claim of the original work that the lack of robustness to refits is a general problem of acceptance-focused recourse more than a problem of a specific learner.
- The extension of the experiments to assess the robustness of the methods to shifts in the data generating process -- albeit at this point rather illustrative -- hints towards an interesting direction for future research.
- Overall, the claims that the authors make are supported with evidence.
- Minor: It is nice to see that the authors also examined the optimization procedure to determine whether it is relevant to the paper's findings.

Weaknesses:
- The new dataset only marginally adds value. The original dataset already includes linear and nonlinear settings; The new dataset adds a low-dimensional example with nonlinearities. It would have been interesting to see an application on a higher-dimensional dataset or to craft one that resembles a real-world dataset.
- The assessment of the robustness to shifts is, at this point, rather illustrative. I would expect that, if the shift in the data is strong enough, ICR recommendations would break down too. Yet, in the experiments, the ICR invalidity rate stays around 0. It would be interesting to have an evaluation with stronger shifts as well (e.g., a figure showing the strength of the perturbation vs the invalidity of recourse). Also, it would be interesting to see what kinds of shifts could break ICR. I get that evaluating this would take up additional computational resources.
- In light of the above points, the wording when describing the conclusions is, in my taste, in a few cases too strong. See, e.g., p8: "ICR is robust not only to refits but also to distribution shifts."
- When describing your methodology, the story is not fully streamlined yet. For example, you describe the hyperparameter settings for the additional robustness experiments in section 3.3 before introducing the robustness assessment in section 3.5. Overall, the writing could be improved. See the detailed comments below for suggestions for improvement.

Overall, the paper is on the borderline to qualify for a reproducibility certification.

---

> ### Author Response · Authors · 2024-04-11
> **Rebuttal**
>
> Thank you for your time and constructive review. We appreciate your recognition of replicating the main findings and the extension experiments. We did our best to facilitate all of the requested changes. We created, like you suggested, a background section where we introduced CE and CR in greater detail and also added a new section where we explained ICR in a better way.
>
> **New dataset adds only marginal value**
>
> We do agree that our added dataset only adds marginal value. We decided to add another small dataset to verify that certain parts of the other datasets were not cherry-picked. The reason for not evaluating the examined methods on a real-life dataset, like the credit dataset, is that these real-life datasets have more variables, and we needed more computational resources.
>
> **Robustness of shifts**
>
> We agree with every point you made about this extension. Indeed, ICR will also break down at one point. We wanted to show the high invalidity for CR and CE, while ICR has meager invalidity rates. The plot you suggested is an exciting idea. We decided not to follow up with this idea because of the extensive computational resources we already used and since using more mean and variance shifts needs to be done for multiple confidence levels, multiple nodes that are shifted, and potentially multiple classifiers to get an unbiased and good comparison. Because we did not implement this, we rephrased our conclusions so as not to sound too strong.
>
> **Writing of the paper**
> In the updated manuscript, a distinct section (Section 5) is devoted to clearly explaining the experiment we reproduced and the ones we conducted additionally for the robustness assessment. Also, a new section (Section 7) summarizes our challenges, how we resolved them, and the communication with the authors. The table regarding the additional experiments on robustness study is now in the same subsection where we give a first overview of our extensions before explaining each extension in detail. Furthermore, we addressed the writing changes you requested.

---

### Review · Reviewer_ugqB · 2024-03-25

**Summary Of Contributions:**

The paper aims to reproduce the main findings of the paper “Improvement-Focused Causal Recourse (ICR)” by Konig et al. The findings of the original paper are largely confirmed. The paper also tests the robustness and generalizability of ICR in new domains not considered in the original paper. The results are overall favorable for ICR.

**Audience:**

No

**Claims And Evidence:**

Yes

**Requested Changes:**

Please make the paper self-contained. One should not have to read Konig et al. to understand the background and main findings.

**Strengths And Weaknesses:**

I applaud the authors for conducting a very detailed reproducibility study. This is something we need more of in the field. However, the current manuscript does not feel like a standalone research paper but more like a technical supplement to the original paper. I would perhaps understand the purpose of this study if the paper of Kong et al. was extremely impactful in the field and we really need to make sure their results are reproducible, but the paper is only very recent and its contributions are not primarily experimental. The robustness tests are not very comprehensive, but the study mainly focused on reproducing the original findings. To make matters worse, the original paper is not explained well in the current manuscript but this very important piece of background is assumed, so the manuscript is not even easily understandable. The fact that knowledge of the ICR work is assumed is one of the factors that make this manuscript feel like a technical supplement. To be clear, it is still valuable to make this study publicly available as an accompanying piece of evidence for the original paper, but right now it feels like it would be of very limited interest.

---

> ### Author Response · Authors · 2024-04-11
> **Rebuttal**
>
> Thank you very much for your time and insight. We appreciated your positive feedback and made some changes to tackle the main issue, related to our paper being more self-contained.
>
> We created a background section introducing CE and CR in greater detail and added a new section explaining the main concepts behind ICR. We tried to maintain a concise style and did not provide all of the equations from the main paper because the supporting explanations would result in a complete reiteration of the original work. We hope to have struck a balance between the two.
>
> In the updated manuscript, a distinct section (Section 5) is devoted to clearly explaining the experiment we reproduced and the ones we conducted additionally for the robustness assessment. Also, a new section (Section 7) summarizes the challenges we faced, how we resolved them, and the communication with the authors.
>
> Regarding robustness tests, we conducted extensive experiments modeled after the methodology proposed in Upadhyay et al. (Towards robust and reliable algorithmic recourse, 2021), and Rawal et al. (Algorithmic recourse in the wild: Understanding the impact of data and model shifts, 2020), but to the extent that our computational resources allowed.

---

### Review · Reviewer_Leaj · 2024-03-29

**Summary Of Contributions:**

This paper reproduced the results by König et al. (2023) that proposed the improvement-focused causal recourse (ICR), confirming all the proposed claims. In addition, they assessed ICR's generalization and robustness against different models and datasets. They also used several genetic algorithms as a optimization strategy to repeat the previous experiments, achieving comparable performance.

**Audience:**

Yes

**Claims And Evidence:**

Yes

**Requested Changes:**

- Make a more detailed comparison among the CE, CR, and ICR with intuitive examples and mathematical definations.
- Clarify the setting's reasonability.
- Add a analysis about the choice of  $\bar{\gamma}$.
- Provide some insightful understanding for current experimental results.

**Strengths And Weaknesses:**

Strengths:
- The reproducible experiment and newly added one in this paper are solid and clearly exhibited. They well recurred the claimed conclusions in König et al. (2023) and made broader empirical analysis.
- The relationship with the previous work is clearly claimed.

Weaknesses:
- The reasonability of two settings is not clear. For section 3.1.1 (Individualized Improvement Confidence), how to obtain ground-truth/reliable SCMs in practical? In addtion, is a SCM shared by a population or personalized? Please provide examples to demonstrate the reasonability of this setting. Maybe I have not totally understand this problem, it seems that the setting of section 3.1.2 is more reasonable.
- There is a lack of experiments on real datasets, such as bank credit or public policy.
- I agree that the authors have done sufficient experimental analysis. However, there is a lack of the corresponding understanding or explaination for the results of generalization and robustness.
- How to choose the $\bar{\gamma}$ in Eq. (1)? Whether there exists a considerable tradeoff between the improvement confidence and the difficult of optimization?

---

> ### Author Response · Authors · 2024-04-11
> **Rebuttal**
>
> Thank you very much for your time and insightful review. We appreciate your positive feedback on how we extended the paper. We are going to address each of your suggested points:
>
> **Reasonability of the settings**
>
> We realized we had not explained these settings in detail enough and had changed the part in the paper. Here are the reasons as well: An SCM must come from an expert in the field. If the SCM is not very reliant, it is better to use only the causal graph (Karimi et al. 2020: "Algorithmic recourse under imperfect causal knowledge: a probabilistic approach"). For the subpopulation approach: The whole population uses the same SCM. How a subpopulation is selected: If you perform actions on some features, you choose individuals that share values for variables that are not descendants of the intervened-upon variables. The subpopulation approach, while performing worse, might be applicable in more settings due to the lack of SCMs.
>
> **Lack on real-life datasets**
>
> Our added dataset only adds marginal value. We decided to add another small dataset to verify that certain parts of the other datasets were not cherry-picked. The reason for not evaluating the examined methods on a real-life dataset, like the credit dataset, is that these real-life datasets have more variables, and we had limited computational resources.
>
> **More insight on the results**
>
> In section 6.2, we added reasoning for why our extensions are important: For classifiers: A method should be able to perform recourse well independently of the classifier used, as in real-life scenarios for different problems, different classifiers are applied. For the data shifts: Data distributions can change over time; if a method can handle shifts in the distribution, it is very applicable as it does not invalidate the results from the data of the old distribution, which is essential in a case where doing an action takes a few years to accomplish (for example getting a degree to get a job). Furthermore, we added the most significant limitation of ICR: the need for a causal graph or SCM.
>
> **Trade-off between confidence score and optimization difficulty**
>
> We explained better in the paper the meaning of $\bar{\gamma}$. In the data, we have the costs for the actions. We do not observe a clear correlation between costs and confidence levels. Regarding the runtime, the genetic algorithm will run for the specified number of generations (see Appendix B) before picking the best result within all generations, regardless of the specified confidence score $\bar{\gamma}$.

---

### Decision · Action_Editor_CRCm · 2024-05-15

**Recommendation:** Reject

**Comment:**

The extensions made to the original experiments appear to be marginal, and the reported results primarily confirm the findings of the original paper without providing significant new insights. Overall, merely downloading simulation code and rerunning small-scale experiments with synthetic data for a theoretical paper does not constitute a substantial contribution.

**Audience:**

Very few people will be interested in the results. Reading the original paper is good enough.

**Claims And Evidence:**

No, the work is simply reproducing the paper with only extension to some more datasets. It does not provide meaningful new findings that could benefit the community.